# Human-In-The-Loop Task and Motion Planning
# for Imitation Learning

**Abstract:** Imitation learning from human demonstrations can teach robots complex manipulation skills, but is time-consuming and labor intensive. In contrast, Task and Motion Planning (TAMP) systems are automated and excel at solving long-horizon tasks, but they are difficult to apply to contact-rich tasks. In this paper, we present Human-in-the-Loop Task and Motion Planning (HITL-TAMP), a novel system that leverages the benefits of both approaches. The system employs a TAMP-gated control mechanism, which selectively gives and takes control to and from a human teleoperator. This enables the human teleoperator to manage a fleet of robots, maximizing data collection efficiency. The collected human data is then combined with an imitation learning framework to train a TAMP-gated policy, leading to superior performance compared to training on full task demonstrations. We compared HITL-TAMP to conventional teleoperation system — users gathered more than 3x the number of demos given the same time budget. Furthermore, proficient agents (75%+ success) could be trained from just 10 minutes of non-expert teleoperation data. Finally, we collected 2.1K demos with HITL-TAMP across 12 contact-rich, long-horizon tasks and show that the system often produces near-perfect agents. Videos and additional results at https://sites.google.com/view/corl-2023-hitl-tamp.

**Keywords:** Imitation Learning, Task and Motion Planning, Teleoperation

## 1 Introduction

Learning from human demonstrations has emerged as a promising way to teach robots complex manipulation skills [1, 2]. However, scaling up this paradigm to real-world long-horizon tasks has been difficult — providing long manipulation demonstrations is time-consuming and labor intensive [3]. At the same time, not all parts of a task are equally challenging. For example, significant portions of complex manipulation tasks such as part assembly or making a cup of coffee are free-space motion and object transportation, which can be readily automated by non-learning approaches such as motion planning. However, planning methods generally require accurate dynamics models [4] and precise perception, which are often unavailable, limiting their effectiveness at contact-rich and low-tolerance manipulation. In this context, our work aims at solving real-world long-horizon manipulation tasks by combining the benefits of learning and planning approaches.

Our method focuses on augmenting Task and Motion Planning (TAMP) systems, which have been shown to be remarkable at solving long-horizon problems [5]. TAMP methods can plan behavior for a wide range of multi-step manipulation tasks by searching over valid combinations of a small number of primitive skills. Traditionally, each skill is hand-engineered; however, certain skills, such as closing a spring-loaded lid or inserting a rod into a hole, are prohibitively difficult to model in a productive manner. Instead, we use a combination of human teleoperation and closed-loop learning to implement just these select skills, keeping the rest automated. These skills use human teleoperation at data collection time and a policy trained from the data at deployment time. Integrating TAMP

Submitted to the 7th Conference on Robot Learning (CoRL 2023). Do not distribute.

systems and human teleoperation poses key technical challenges — special care must be taken to enable seamless handoff between them to ensure efficient use of human time.

To address these challenges, we introduce Human-in-the-Loop Task and Motion Planning (HITL-TAMP), a system that symbiotically combines TAMP with teleoperation. The system collects demonstrations by employing a TAMP-gated control mechanism — it trades off control between a TAMP system and a human teleoperator, who takes over to fill in gaps that TAMP delegates. Critically, human operators only need to engage at selected steps of a task plan when prompted by the TAMP system, meaning that they can manage a fleet of robots by asynchronously engaging with one demonstration session at a time while a TAMP system controls the rest of the fleet.

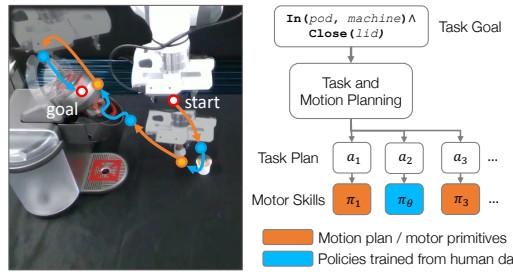

Figure 1: **Overview.** HITL-TAMP decomposes a task (making coffee) into planning-based (TAMP) and learning-based (human imitation) segments.

By soliciting human demonstrations only when needed, and allowing for a human to participate in multiple parallel sessions, our system greatly increases the throughput of data collection while lowering the effort needed to collect large datasets on long-horizon, contact-rich tasks. We combine our data collection system with an imitation learning framework that trains a TAMP-gated policy (as illustrated in Fig. 1) on the collected human data. We show that this leads to superior performance compared to collecting human demonstrations on the entire task, in terms of the amount of data and time needed for a human to teach a task to the robot, and the success rate of learned policies.

**The main contributions of this paper are:**
• We develop HITL-TAMP, an efficient data collection system for long-horizon manipulation tasks that synergistically combines and trades off control between a TAMP system and a human operator.
• HITL-TAMP contains novel components including (1) a mechanism that allows TAMP to learn planning conditions from a small number of demonstrations and (2) a queuing system that allows a demonstrator to manage a fleet of parallel data collection sessions.
• We conduct a study (15 users) to compare HITL-TAMP with a conventional teleoperation system. Users collected over 3x more demos with our system given the same time budget. Proficient agents (over 75% success) could be trained from just 10 minutes of non-expert teleoperation data.
• We collected 2.1K demos with HITL-TAMP across 12 contact-rich and long-horizon tasks, including real-world coffee preparation, and show that HITL-TAMP often produces near-perfect agents.

## 2   Preliminaries

**Summary of Related Work.** Several works have shown the value in learning robot manipulation with human demonstrations [6, 1, 2, 7, 8, 9, 10, 11, 8], in developing automatic control hand-offs between a human supervisor and an automated system for more effective data collection [12, 13, 14, 15, 16], and in combining learned and predefined skills [17, 18, 19, 20, 21]. Prior TAMP [5, 22, 4, 23] works have also integrated learning-based components [24, 25, 26, 27, 28, 29, 30, 31, 32] to make less assumptions on prior knowledge. **See Appendix D for full related work.**

**Problem Statement.** We consider a robot acting in a discrete-time Markov Decision Process (MDP) $\langle \mathcal{X}, \mathcal{U}, \mathcal{T}(x' \mid x, u), \mathcal{R}(x), \mathcal{P}_0 \rangle$ defined by state space $\mathcal{X}$, action space $\mathcal{U}$, transition distribution $\mathcal{T}$, reward function $\mathcal{R}$, and initial state distribution $\mathcal{P}_0$. We assume we are given an offline dataset of $N$ partial demonstration trajectories (collected via our HITL-TAMP system, see Sec. 3.3) $\mathcal{D} = \{\langle x_0^i, u_0^i \rangle, \langle x_1^i, u_1^i \rangle, ..., x_{T^i}^i\}_{i=1}^N$. We train policies $\pi$ with Behavioral Cloning [33] using the objective $\arg\min_\theta \mathbb{E}_{(x,u) \in \mathcal{D}} ||\pi_\theta(x) - u||^2$ (details in Appendix K).

We consider a TAMP policy $\pi_t(u \mid x)$ for controlling the robot. It plans a sequence of actions that will be tracked using a feedback controller. We use the PDDLStream [23] planning framework, a

logic-based action language that supports planning with continuous values, to model our TAMP domain. States and actions are described using *predicates*, Boolean functions, which can have discrete and continuous parameters. A predicate paired with values for its parameters is called a *literal*. Our TAMP domain uses the following parameters: $o$ is an object, $g \in \mathrm{SE}(3)$ is a 6-DoF object grasp pose relative to the gripper, $p \in \mathrm{SE}(3)$ is an object placement pose, $q \in \mathbf{R}^d$ is a robot configuration with $d$ DoFs, and $\tau$ is a robot trajectory comprised of a sequence of robot configurations.

The planning state $s$ is a set of true literals for *fluent* predicates, predicates who's truth value can change over time. We define the following fluent predicates: $\mathrm{AtPose}(o, p)$ is true when object $o$ is placed at placement $p$; $\mathrm{AtGrasp}(o, g)$ is true when object $o$ is grasped using grasp $g$; $\mathrm{AtConf}(q)$ is true when the robot is at configuration $q$; $\mathrm{Empty}()$ is true when the robot's end effector is empty; $\mathrm{Attached}(o, o')$ is true when object $o$ is attached to object $o'$;

We use the **Tool Hang** task as a running example (see Fig. 5), where the robot must insert a *frame* into a *stand* and then hang a *tool* on the *frame*. The set of goal system states $\mathcal{X}_*$ is expressed as a *logical formula* over literals. Let $s_0$ be the initial state $s_0$ and $G$ be the goal formula:

$$s_0 = \{\mathrm{AtPose}(frame, \boldsymbol{p_0^f}), \mathrm{AtPose}(tool, \boldsymbol{p_0^t}), \qquad G = \mathrm{Attached}(frame, stand) \, \wedge$$
$$\mathrm{AtPose}(stand, \boldsymbol{p_0^s}), \mathrm{AtConf}(\boldsymbol{q_0}), \mathrm{Empty}()\}. \qquad \mathrm{Attached}(tool, frame) \, \wedge \, \mathrm{Empty}().$$

Planning actions $a$ are represented using action schemata. An action schema is defined by a 1) name, 2) list of parameters, 3) list of *static* (non-fluent) literal *constraints* (**con**) that valid parameter values satisfy, 4) list of fluent literal *preconditions* (**pre**) that must hold to correctly execute the action, and 4) list of fluent literal *effects* (**eff**) that specify changes to state. The move action advances the robot from configuration $q_1$ to $q_2$ via trajectory $\tau$. The constraint $\mathrm{Motion}(q_1, \tau, q_2)$ is satisfied if $q_1$ and $q_2$ are the start and end of $\tau$. In the pick action, the constraint $\mathrm{Grasp}(o, g)$ holds if $g$ is a valid grasp for object $o$, and the constraint $\mathrm{Pose}(o, p)$ holds if $p$ is a valid placement for object $o$. The explicit constraint $f(q) * g = p$ represents kinematics, namely that forward kinematics $f : \mathbf{R}^d \to \mathrm{SE}(3)$ for the robot's gripper from configuration $q$ multiplied with grasp $g$ produces pose $p$.

```
move(q₁, τ, q₂)                        pick(o, g, p, q)
  con: [Motion(q₁, τ, q₂)]               con: [Grasp(o, g), Pose(o, p), [f(q) * g = p]]
  pre: [AtConf(q₁), Safe(τ)]             pre: [AtPose(o, p), Empty(), AtConf(q)]
  eff: [AtConf(q₂), ¬AtConf(q₁)]         eff: [AtGrasp(o, g), ¬AtPose(o, p), ¬Empty()]
```

The limitations of the TAMP system are that, although it can readily observe the robot state, it does not have the ability to precisely estimate the environment and productively react to changes in it in real-time. Thus, it's advantageous to teleoperate skills that require 1) contact-rich interaction that is difficult to accurately model and 2) precision greater than that which the perception system can deliver. An example of 1) is the insertion phase of **Tool Hang**, which typically requires contacting the walls of the hole to align the *frame*, and an example of 2) is the hanging phase of **Tool Hang**, which requires precisely aligning the hole of the tool with the resting *frame*.

## 3 Integrating Human Teleoperation and TAMP

To make TAMP and conventional human teleoperation systems compatible, we describe crucial components that allow for seamless handoff between TAMP and a human operator. These include 1) a novel constraint learning mechanism that allows TAMP to plan to states that enable subsequent human teleoperation (Sec. 3.2) and 2) the core TAMP-gated teleoperation algorithm (Sec. 3.3).

### 3.1 Teleoperation Action Modeling

To account for human teleoperation during planning, we need an approximate model of the teleoperation process. We build on the high-level modeling approach of Wang *et al.* [25] by specifying an action schema for each skill identifying which constraints can be modeled using classical techniques. Then, we extract the remaining constraints from a handful of teleoperation trajectories. Continuing our running example, we teleoperate the *frame* insertion and *tool* hang in the **Tool Hang** task.

The `attach` action models any skill that involves attaching one movable object to another object, for example, by placing, inserting, or hanging. Its parameters are a held object $o$, the current grasp $g$ for $o$, the corresponding current pose $p$ of $o$, the current robot configuration $q$, the subsequent pose $\widehat{p}$ of $o$, the subsequent robot configuration $\widehat{q}$, and the object to be attached to $o'$. This action is stochastic as the human teleoperator "chooses" the resulting pose $\widehat{p}$ and configuration $\widehat{q}$ (indicated by $\widehat{\Box}$), which modeled by the constraint $\texttt{HumanAttach}(o, \widehat{p}, \widehat{q}, o')$. Rather than explicitly model this constraint, we take an *optimistic* determinization of the outcome by assuming that the human produces a satisficing $\widehat{p}, \widehat{q}$ pair, without committing to specific numeric values.

$\texttt{attach}(o, g, p, q, \widehat{p}, \widehat{q}, o')$
   **con:** $[\texttt{AttachGrasp}(o, g), \ \underline{\texttt{PreAttach}(o, p, o')}, \ [f(q) * g = p],$
         $\underline{\texttt{GoodAttach}(o, \widehat{p}, o')}, \ \underline{\texttt{HumanAttach}(o, \widehat{p}, \widehat{q}, o')}]$
   **pre:** $[\texttt{AtGrasp}(o, g), \ \texttt{AtConf}(q)]$
   **eff:** $[\texttt{AtPose}(o, \widehat{p}), \ \texttt{Empty}(), \ \texttt{Attached}(o, o'), \ \texttt{AtConf}(\widehat{q}), \ \neg\texttt{AtGrasp}(o, g), \ \neg\texttt{AtConf}(q)]$

The key constraint is $\texttt{GoodAttach}(o, \widehat{p}, o')$, which is true if object $o$ at pose $p$ satisfies the ground-truth goal attachment condition in $\mathcal{G}$ with object $o'$. The human teleoperator is tasked with reaching a pose $\widehat{p}$ that satisfies this constraint, which is a postcondition of the action. The goal of model learning is to represent the preconditions (Sec. 3.2) that facilitate this in a generative fashion.

## 3.2 Constraint Learning

To complete the action model, we learn the $\texttt{AttachGrasp}$ and $\texttt{PreAttach}$ constraints, which involve parameters in $\texttt{attach}$'s preconditions. We bootstrap these constraint models from a few ($\sim 3$ in our setting) human demonstrations. These demonstrations only need to showcase the involved action. Through compositionality, these actions can be deployed in many new tasks without the need for retraining.

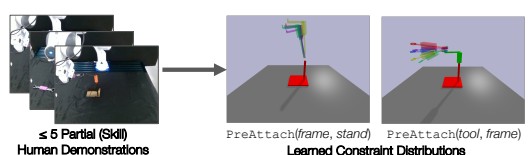

≤ 5 Partial (Skill) Human Demonstrations     PreAttach(*frame, stand*)    PreAttach(*tool, frame*)
Learned Constraint Distributions

Figure 2: **Constraint learning.** Example of learned attach conditions for the *frame* (*left*) and *tool* from a handful of demonstrations for the **Tool Hang** task.

In this work, because the set of objects is fixed, the constraints do not need to generalize across objects so we simply populate uniform distributions over poses conditioned on task and objects. In settings where there are novel objects at test time, we could instead estimate these affordances across objects directly from observations [25, 34, 35] using more complicated (deep) generative models.

We define $\texttt{PreAttach}(o, p, o')$ to be true if $p$ is a pose for object $o$ immediately prior to the human achieving $\texttt{GoodAttach}(o, \widehat{p}, o')$. For each human demonstration, we start at the first state where $\texttt{GoodAttach}$ is satisfied and then search backward in time for the first state where (1) the robot is holding object $o$ and (2) objects $o$ and $o'$ are at least $\delta$ centimeters apart. This minimum distance constraint ensures that $o$ and $o'$ are not in contact in a manner that is spatially consistent and robust to perception and control error. We log the relative pose $p$ between $o$ and $o'$ as a data point and continue iterating over human demonstrations to populate a dataset $P_{o'}^o = \{p \mid \texttt{PreAttach}(o, p, o')\}$.

Similarly, we define $\texttt{AttachGrasp}(o, g)$ to be true if $g$ is a grasp for object $o$ allows for the human achieving $\texttt{GoodAttach}(o, \widehat{p}, o')$. Not all object grasps enable the human to satisfy the target condition, for example, a *frame* grasp on the tip that needs to be inserted. Similar to $\texttt{PreAttach}$, for each demonstration we log the relative pose between the robot end effector and object $o$ at the first pre-contact state before satisfying $\texttt{GoodAttach}$, producing dataset $G^o = \{g \mid \texttt{AttachGrasp}(o, g)\}$.

## 3.3 TAMP-Gated Teleoperation

We now describe TAMP-gated teleoperation, where a TAMP system decides when to execute portions of a task, and when a human operator should complete a portion (full details in Appendix J). Each teleoperation episode consists of one or more *handoffs* where the TAMP system prompts a human operator to control a portion of a task, or where the TAMP system takes control back after it determines that the human has completed their segment.

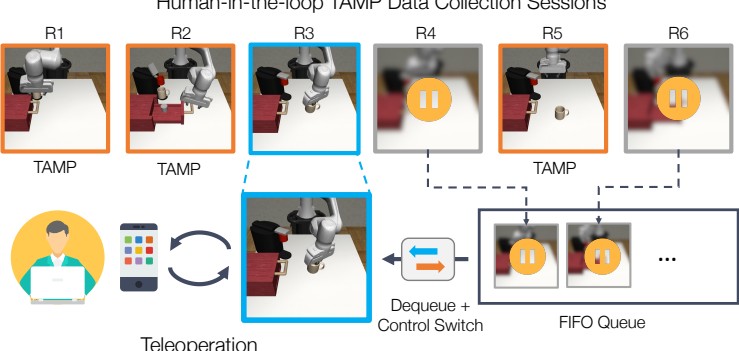

Figure 3: **Queueing system.** HITL-TAMP's queueing system allows a human teleoperator (bottom left) to manage a fleet of asynchronously-running data collection sessions (R1-R6).

Every task is defined by a goal formula $G$. On each TAMP iteration, it observes the current state $s$. If it satisfies $G$ the episode terminates, otherwise, the TAMP system solves for a plan $\vec{a}$ from current state $s$ to the goal $G$. TAMP subsequently issues joint position commands to carry out planned motions until reaching an action $a$ requiring the human. Next, control switches into teleoperation mode, where the human has full 6-DoF control of the end effector. We use a smartphone interface similar to prior teleoperation systems [36, 37, 10]. The robot end effector is controlled using an Operational Space Controller [38]. The TAMP system monitors whether the state satisfies the planned action postconditions $a.\textit{effects}$. Once satisfied, control switches back to the TAMP system, which replans.

## 4   Scaling Data Collection for Learning

**Increasing Data Throughput with a Queueing System.** Since the TAMP system only requires human assistance in small parts of an episode, a human operator has the opportunity to manage multiple robots and data collection sessions simultaneously. To this end, we propose a novel queueing system (Fig. 3) allowing each operator to interact with a fleet of robots. We implement this by using several ($N_{\text{robot}}$) **robot processes**, a single **human process**, and a **queue** (more analysis in Appendix I). Each **robot process** runs asynchronously, and spends its time in 1 of 3 modes — (1) being controlled by the TAMP system, (2) waiting for human control, or (3) being controlled by the human. This allows the TAMP system to operate multiple robots in parallel. When the TAMP system wants to prompt the human for control, it enqueues the environment into the shared queue. The **human process** communicates with the human teleoperation device and sends control commands to one robot process at a time. When the human completes a segment, TAMP resumes control of the robot, and the human process dequeues the next session from the queue.

**TAMP-Gated Policy Deployment.** HITL-TAMP results in demonstrations that consist of TAMP-controlled parts and human-controlled parts — we train a policy with Behavioral Cloning [33] on the human portions (details in Appendix K). To deploy the learned agent, we use a TAMP-gated control loop that is identical to the handoff logic in Sec. 3.3, using the policy instead of the human.

## 5   Experiment Setup

**Tasks.** We chose evaluation tasks that are *contact-rich* and *long-horizon*, to validate that HITL-TAMP indeed combines the benefits of the two paradigms (see Fig. 4 and Fig. 5). We further evaluated HITL-TAMP on variants of tasks where objects are initialized in **broad** regions of the workspace, a difficult setting for imitation learning systems in the past. Full details in Appendix E.

**Pilot User Study.** We conducted a pilot user study with 15 participants to compare our system (HITL-TAMP) to a conventional teleoperation system [36], where task demonstrations were collected without TAMP involvement. Each participant performed task demonstrations on 3 tasks (**Coffee**, **Square (Broad)**, and **Three Piece Assembly (Broad)**) for 10 minutes on each system,

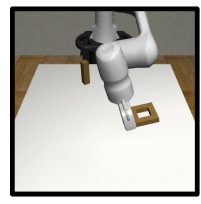 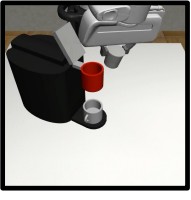 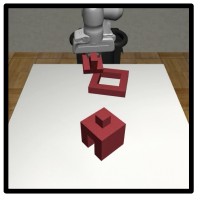 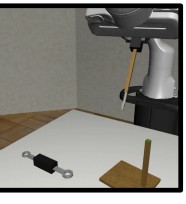 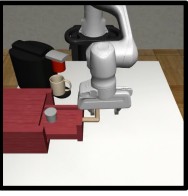

| (a) Square | (b) Coffee | (c) 3 Pc. Assembly | (d) Tool Hang | (e) Coffee Prep. |

Figure 4: **Tasks.** We use HITL-TAMP to collect demonstrations for contact-rich, long-horizon tasks.

| Task | Demos (avg-user) | Demos (avg-novice) | Demos (all) | SR (avg-user) | SR (avg-novice) | SR (all) |
|---|---|---|---|---|---|---|
| Coffee (C) | 11.2 | 7.2 | 168.0 | 24.4 | 15.0 | 76.0 |
| Coffee (HT) | **28.7** | **25.2** | **431.0** | **90.7** | **90.0** | **100.0** |
| Square Broad (C) | 11.1 | 5.2 | 166.0 | 1.2 | 0.0 | 20.0 |
| Square Broad (HT) | **49.8** | **41.8** | **747.0** | **80.0** | **77.5** | **98.0** |
| Three Piece Assembly Broad (C) | 7.8 | **7.0** | 117.0 | 0.0 | 0.0 | 0.0 |
| Three Piece Assembly Broad (HT) | **15.1** | 8.0 | **227.0** | **27.7** | **17.5** | **66.0** |

Table 1: **User Study Data Collection and Policy Learning Results.** We report the number of demos collected averaged across users (avg-user), averaged across novice users (avg-novice), and summed across all users (all). We also report the success rate of policies trained on per-user data (avg-user: averaged across all users, and avg-novice: averaged across novice users), and trained on all user data (all). Users collected more demonstrations using HITL-TAMP (HT) than the conventional system (C), and policy performance was vastly greater as well.

totaling 60 minutes of data collection across the 3 tasks and 2 systems. Participants filled out a post-study survey to rank their experience with both systems. Each participant's number of successful demonstrations was recorded to evaluate the data throughput of each system, and agents were trained on each participant's demonstrations and across all participants' demonstrations (Sec. 6.1).

## 6 Experiment Results

We (1) present user study results to highlight HITL-TAMP's data collection efficiency (Sec. 6.1), (2) compare trained HITL-TAMP agents to policies trained from full task demonstrations (Sec. 6.2), and (3) deploy HITL-TAMP in the real world without precise perception (Sec. 6.3).

### 6.1 System Evaluation: User Study

We show that (1) HITL-TAMP allows participants to collect demonstrations much faster than conventional teleoperation, (2) we can train performant policies using data collected from users with varying system proficiency, (3) HITL-TAMP enables novice operators to collect high-quality demonstration data, and (4) HITL-TAMP requires less user effort than conventional teleoperation.

**HITL-TAMP enables users to collect task demonstrations at a much higher rate than a conventional teleoperation system.** As Table 1 shows, collectively, our 15 users gathered 2.5x more demonstrations with HITL-TAMP when compared to the conventional system on the Coffee task (431 vs. 168), 4.5x more on Square Broad (747 vs. 166), and nearly 2x more on Three Piece Assembly Broad (227 vs. 117). The high collection efficacy of HITL-TAMP was also reflected on a per-user basis — users averaged 28.7 demos on Coffee (vs. 11.2), 49.8 demos on Square Broad (vs. 11.1), and 15.1 demos on Three Piece Assembly Broad (vs. 7.8), during their 10-minute sessions.

**HITL-TAMP enables performant policies to be trained from minutes of data.** We used each person's 10-minute demonstrations to train a policy for each (user-task) pair with behavioral cloning. Agents trained on HITL-TAMP data vastly outperformed those trained from the conventional teleoperation data (Table 1) — agents achieved an average success rate of 90.7% on Coffee (vs. 24.4%), 80.0% on Square Broad (vs. 1.2%), and 27.7% on Three Piece Assembly Broad (vs. 0.0%).

**HITL-TAMP enables training proficient agents from multi-user data.** Prior work [39, 1] noted that imitation learning from multi-user demonstrations can be difficult. However, we found agents trained on the full set of multi-user HITL-TAMP data achieve high success rates (100.0%, 98.0%, and 66.0% on Coffee, Square Broad, and Three Piece Assembly Broad, respectively) compared to

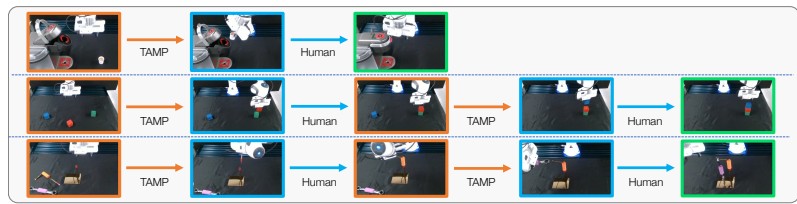

| Task | Success Rate |
|------|--------------|
| Stack Three | 62.0 |
| Coffee | 74.0 |
| Coffee Broad | 66.0 |
| Tool Hang | 64.0 |

Figure 5: (left) **Real Tasks.** Coffee (*top*), a version where the machine can be on either side (Coffee Broad, not shown), Stack Three (*middle*), and **Tool Hang** (*bottom*). We show TAMP in orange and the human in blue. (right) **Real World Policy Performance.** We collected 100 demonstrations on Stack Three, Coffee, and Coffee Broad and 50 demonstrations on Tool Hang with HITL-TAMP and report the policy performance in this table.

| Task | Time (min) | SR (low-dim) | SR (image) |
|------|-----------|--------------|------------|
| Square | 13.5 | 100.0 ± 0.0 | 100.0 ± 0.0 |
| Square Broad | 14.0 | 100.0 ± 0.0 | 100.0 ± 0.0 |
| Coffee | 22.6 | 100.0 ± 0.0 | 100.0 ± 0.0 |
| Coffee Broad | 28.8 | 99.3 ± 0.9 | 96.7 ± 0.9 |
| Tool Hang | 48.0 | 80.7 ± 1.9 | 78.7 ± 0.9 |
| Tool Hang Broad | 51.5 | 49.3 ± 1.9 | 40.7 ± 0.9 |
| Three Piece Assembly | 30.0 | 100.0 ± 0.0 | 100.0 ± 0.0 |
| Three Piece Assembly Broad | 34.9 | 84.7 ± 4.1 | 82.0 ± 1.6 |
| Coffee Preparation | 78.4 | 96.0 ± 3.3 | 100.0 ± 0.0 |

| Task | Time (min) | SR (im) | TAMP-gated SR (im) |
|------|-----------|---------|--------------------|
| Square (C) [1] | 25.0 | 82.0 ± 0.0 | 100.0 ± 0.0 |
| Square (HT) | **13.5** | **100.0 ± 0.0** | **100.0 ± 0.0** |
| Square Broad (C) | 48.0 | 15.3 ± 0.0 | 94.7 ± 0.9 |
| Square Broad (HT) | **14.0** | **100.0 ± 0.0** | **100.0 ± 0.0** |
| Three Piece Assembly (C) | 60.0 | 75.3 ± 0.0 | 77.3 ± 7.7 |
| Three Piece Assembly (HT) | **30.0** | **100.0 ± 0.0** | **100.0 ± 0.0** |
| Tool Hang (C) [1] | 80.0 | 67.3 ± 0.0 | **82.0 ± 2.8** |
| Tool Hang (HT) | **48.0** | **78.7 ± 0.9** | 78.7 ± 0.9 |

Figure 6: (left) **Results on HITL-TAMP datasets.** We collected 200 demonstrations on each task with HITL-TAMP and trained low-dim and visuomotor agents TAMP-gated agents on each dataset. (right) **Comparison to conventional teleoperation datasets.** We trained both normal and TAMP-gated policies using conventional teleoperation (C) and compared them to HITL-TAMP (HT). Surprisingly, TAMP-gating makes policies trained on the data comparable to HITL-TAMP data, but data collection still involves significantly higher operator time.

those trained on the full set of conventional teleoperation data (76.0%, 20.0%, 0.0%) (see Table 1). In fact, the worst per-user HITL-TAMP policy (10-minutes of data) outperformed the policy trained on the full set of conventional teleoperation data (150 minutes) on both Square Broad (56.0% vs. 20.0%) and Three Piece Assembly Broad (14.0% vs. 0.0%).

**HITL-TAMP enables non-experts to demonstrate tasks efficiently.** 4 of the 15 users in our study had no experience with teleoperation. Table 1 shows we found that they were able to collect far more data on average with HITL-TAMP (more than 3x on Coffee, more than 8x on Square Broad) and policies trained on their HITL-TAMP data achieved significantly higher success over the conventional system — 90.0% (vs. 15.0%) on Coffee, 77.5% (vs. 0.0%) on Square Broad, and 17.5% (vs. 0.0%) on Three Piece Assembly Broad.

**HITL-TAMP results in a lower perceived workload compared to the conventional teleoperation system.** Each participant completed a NASA-TLX survey [40] to rank their perceived workload for each system across 6-categories (100-point scale, increments of 5). Users found HITL-TAMP to require less mental demand (36% vs. 74%), less physical demand (29.7% vs. 63.7%), and less temporal demand (28.3% vs. 53.7%), while enabling higher overall performance (83.7% vs. 59.7%), with lower effort (29.3% vs. 75.7%) and lower frustration (30.0% vs. 65.0%).

## 6.2 Learning Results

We collect datasets with HITL-TAMP across 9 tasks (see Sec. 5) and show that highly capable policies can be trained from this data. The results compare favorably to training on equal amounts of demonstrations from a conventional teleoperation system.

**HITL-TAMP is broadly applicable to a wide range of contact-rich and long-horizon tasks.** Using HITL-TAMP, we had a single human operator collect 200 demonstrations on each of our tasks. We then trained agents from this data on two observation spaces — *low-dim* observations, where agents directly observe the poses of relevant objects, and *image* observations, where agents observe a front-view RGB image and wrist RGB image (as in [1]). Table 6 shows that across both observation spaces, HITL-TAMP trains near-perfect agents on several tasks (Square, Coffee, Three Piece Assembly), including **broad** tasks with a wide distribution of object initialization (Square Broad, Coffee Broad, Three Piece Assembly Broad). HITL-TAMP also achieves high performance on the Tool Hang task (80.7% low-dim, 78.7% image), which is the hardest task in the robomimic

benchmark [1]. It is also able to train performant agents (49.3% low-dim, 40.7% image) on a **broad** version of the task (Tool Hang Broad). Finally, HITL-TAMP trains near-perfect agents (96%) on the Coffee Preparation task, which consists of several stages (4 TAMP segments and 4 policy segments) involving low-tolerance mug placement, drawer grasping and opening, lid opening, and pod insertion and lid closing.

**HITL-TAMP compares favorably to conventional teleoperation systems in terms of operator time and policy learning.** Even when an equal number of task demonstrations are used, learned policies from HITL-TAMP still outperform those from conventional teleoperation. We run our comparison on 4 tasks — Square, Square Broad, Three Piece Assembly, and Tool Hang, where each task has 200 HITL-TAMP demos collected and 200 conventional system demos. As Table 6 shows, HITL-TAMP enabled collecting 200 demonstrations on each task in much shorter periods of time (additional analysis in Appendix F). Furthermore, agents trained on HITL-TAMP data outperform agents trained on conventional data (with the largest gap being 100.0% vs. 15.3% on Square Broad).

**TAMP-gated control is a crucial component to train proficient policies.** We took the 200 demonstration datasets collected via conventional teleoperation, trained the agents as normal, but deployed them with TAMP-gated control during policy evaluation. This dramatically increases their success rates and gives comparable results to HITL-TAMP data (see Table 6). This shows that datasets consisting of entire human demonstration trajectories are compatible with TAMP-gated control. However, they remain time-consuming to collect, and HITL-TAMP greatly reduces the time needed.

### 6.3 Real Robot Validation

We apply HITL-TAMP to a physical robot setup with a robotic arm, a front-view camera, and a wrist-mounted camera. The only significant change from simulation is the need for perception to obtain pose estimates of the objects to populate the TAMP state. We do not assume any capability to track object poses in real-time. Instead, we allow the human to demonstrate (and the policy to imitate) behaviors from partial observations (RGB cameras). We collected 100 demonstrations for each of 3 tasks — Stack Three, Coffee, and Coffee Broad, and 50 demonstrations on Tool Hang, and report policy learning results across 50 evaluations for each task (25 for Tool Hang) (see Fig. 5). Our TAMP-gated agent achieves 62% on Stack Three, 74% on Coffee, 66% on Coffee Broad (72% with the machine on the right side of the table, and 60% with the machine on the left side), and **64% on Tool Hang (as opposed to the 3% from 200 human demonstrations in prior work [1]).**

## 7 Limitations

**See Appendix C for full limitations.** We assume tasks can be described in PDDLStream and that human teleoperators can demonstrate them. The tasks in this work focus on tabletop domains with limited object variety — future work could scale HITL-TAMP to more diverse settings. Currently, HITL-TAMP requires prior information (at a high-level) on which task portions will be difficult for TAMP. We also assume access to coarse object models and approximate pose estimation to conduct TAMP segments in the real world. Future work could relax these assumptions by integrating perception uncertainty estimates, and extending TAMP to not require object models [35].

## 8 Conclusion

We presented a new approach to teach robots complex manipulation skills through a hybrid strategy of automated planning and human control. Our system, HITL-TAMP, collects human demonstrations using a TAMP-gated control mechanism and learns preimage models of human skills. This allows for a human to efficiently supervise a team of worker robots asynchronously. The combination of TAMP and teleoperation in HITL-TAMP results in improved data collection and policy learning efficiency compared to collecting human demonstrations on the entire task.

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

# Appendix

## A  Table of Contents

## B Frequently Asked Questions (FAQ)

1. **How did you select those specific baselines and ablations in Sec. 6?**

   Our experiments showcase the capabilities of HITL-TAMP as (1) a scalable demonstration collection system and (2) an efficient learning and control framework. To show its value in collecting human demonstrations over an alternative, we compared it extensively against a widely-adopted conventional teleoperation paradigm used in prior works that collect and learn from human demonstrations [1, 6, 2, 7, 41, 20, 9, 36, 37, 42, 10, 15, 16, 11] (see Table 1 and Fig. 6).

   To show its value in learning policies for manipulation tasks, we investigated the value of the core component - the TAMP-gated control mechanism (described in Appendix J). We showed that even policies trained on conventional teleoperation data benefit substantially from incorporating the TAMP-gated control mechanism (Fig. 6). Our TAMP-gated control is a novel control algorithm made possible by key technical components of HITL-TAMP (as described in Sec. 3).

   There are other systems that are designed for specific contact-rich manipulation (such as peg insertion [43, 44]), but HITL-TAMP was not designed to be specialized for any specific task. Rather, it was meant to be a general-purpose system that can be applied to any contact-rich, long-horizon manipulation task, as long as the task can be demonstrated by a human operator, and described in PDDLStream.

2. **How does this work compare with other works that combine imitation learning and TAMP?**

   Prior works, such as [45], trained agents in simulation to imitate demonstration data provided by a TAMP supervisor in simulation. In this way, during deployment, an agent can operate without privileged information (such as object poses) required by TAMP. However, this setting makes a strong assumption that the TAMP system can already solve the target tasks. By contrast, our work extends a TAMP system's capabilities using an agent trained on human demonstration segments collected by HITL-TAMP (training details in Appendix K) in order to solve complex contact-rich tasks in the real world. Training an agent on the TAMP segments collected by HITL-TAMP in order to enable TAMP-free policy deployments is an exciting application for future work. However, it is orthogonal to the main contributions in this paper.

3. **What are the trade-offs between the effort to provide demos and the effort to design models and controllers used in TAMP?**

   Collecting a large number of human demos can be labor and time intensive [11, 7, 37], but extensive modeling of a task for TAMP can similarly be time-consuming. Our system achieves a good tradeoff, by lessening the modeling burden for TAMP by deferring difficult task segments to the human, and lessening the human operator burden by only asking them to operate small segments of a task. When deploying HITL-TAMP (especially in real-world settings), there is significant flexibility in deciding what information is available to the TAMP system in order to automate portions of a task, and which portions of a task should instead be deferred to a human operator (or trained agent).

4. **How does the TAMP system determine which parts of a task plan require a human operator?**

   We formalize human-teleoperated TAMP skills in Sec. 3.1. While their discrete structure is provided by a human (e.g. which objects are involved), our novel action constraint learning technique (Sec. 3.2) characterizes their continuous action parameters. Human modelers have flexibility in deciding which skills should be teleoperated based on the contact-richness and required precision of the interaction. In our experiments, we used a prior understanding of the TAMP system and the limits of planners and perception to determine which parts would require human teleoperation. Other practical alternatives include using uncertainty estimates from perception, or directly applying TAMP to tasks of interest, and

observing sections of failure. Fig. E.1 (in Appendix E) showcases the parts of each task that are handled by the TAMP system and the parts that are handled by the human (or trained agent).

5. **What assumptions are needed to apply HITL-TAMP to real-world settings, as opposed to simulation?**

   Typically, TAMP systems place a high burden on real-world perception, as accurate perception and dynamics models are often needed by TAMP for planning. Part of the motivation of our work was to reduce this requirement. While we do assume knowledge of crude object models and the ability to associate objects (see Sec. 6.3), we use a very simple perception pipeline in this work. We show that this simple pipeline suffices, **even for the challenging Tool Hang task in the real-world** since a human or an end-to-end trained policy handles the most challenging, contact-rich interactions. See Appendix G for additional validation that HITL-TAMP can tolerate noisy perception.

6. **Why are some of the settings for the real-world Tool Hang task different from the other real-world tasks?**

   The data collection and policy learning methodology are identical to the other tasks, but there are a few minor differences. We used an increased resolution of 240x240 for the RGB images (instead of 120x120) due to the need for high-precision manipulation. We also excluded the wrist-view in observations provided to the trained agent, since we found that it was completely occluded during the human portions of the task. Finally, we evaluated our agent over 25 episodes (instead of 50 evaluation episodes used for the other tasks), because policy evaluation for this task is significantly more time-consuming) and obtained a task success rate of 64%, along with a frame insertion rate of 88%.

7. **Why are TAMP plans carried out with a joint position controller, while human teleoperation and learned policies use an OSC controller?**

   Our TAMP system creates plans directly in joint space, so we are able to carry out and track motion plans with higher fidelity by using a joint position controller. On the other hand, human teleoperation requires an end effector controller (we use OSC [38]) to provide an intuitive mapping between the user device and robot control. Consequently, we switch between these two controllers depending on whether the TAMP system or the human is operating the robot. See Appendix J for more information.

## C  Limitations

In this section, we discuss some limitations of HITL-TAMP, which future work can address.

1. **Applicable tasks.** Our general-purpose system can be deployed on any tasks that (1) can be described in PDDLStream and (2) human operators can demonstrate. We did not engineer the system for any specific task — our system greatly extends the set of tasks that can be solved when compared to TAMP alone.

2. **Task variety.** The tasks in this work are focused on tabletop domains, and there is limited object variety in each task. Scaling HITL-TAMP to work for more scenes and objects requires a richer set of assets and scenes (in simulation) and a more robust perception pipeline in the real world.

3. **Prior information on what is difficult for TAMP.** HITL-TAMP requires prior information (at a high-level) on which task portions will be difficult for TAMP. Being able to automatically identify when human demonstrations are needed (e.g. based on uncertainty estimates from perception) is left for future work.

4. **Perception for TAMP.** We assume access to coarse object models and approximate pose estimation in order to conduct the TAMP segments. Future work could relax this assumption by integrating TAMP methods that do not require object models [35].

# D Related Work

## D.1 Demonstration Collection Systems for Robot Manipulation

Recent studies have shown the effectiveness of teaching robots manipulation skills through human demonstration [6, 1, 2, 7, 8, 9]. High-quality, large-scale demonstrations are crucial to this success [2]. Although recent advancements have made demonstration collection systems more scalable and user-friendly [6, 36], collecting a substantial amount of high-quality, long-horizon demonstrations remains time-consuming and labor-intensive [2]. On the other hand, intervention-based systems [46, 42, 47, 48] allow the demonstrator to proactively correct for near-failure cases. However, such systems require users to constantly monitor robot task executions, which is equally time-consuming and sometimes more cognitively-demanding than demonstrating a task [49]. Our system uses a TAMP-gated mechanism that automatically switches control between the robot and the demonstrator. The mechanism also enables a user to demonstrate for multiple sessions asynchronously, dramatically increasing the throughput of task demonstration.

A number of recent works have also investigated automatic control hand-offs in the context of online imitation learning [12, 13, 14, 15, 16, 50, 51, 52]. These works have largely focused on iteratively improving a single learned policy, and the gating mechanisms rely on predicting task performances and action uncertainties, which are often policy and data-specific. Our work instead proposes to augment a TAMP system with imitation-learned policies. The symbolic abstractions of the TAMP system readily delineate TAMP's capabilities and can be used to determine the conditions for control hand-offs.

Our HITL-TAMP also acts as a TAMP-assisted teleoperation system. However, unlike most prior works in assisted robot teleoperation, for which the aims are for humans to provide high-level guidance for low-level autonomous control [53, 54, 55], HITL-TAMP focuses on allowing human teleoperators to "fill the gap" for a TAMP system to complete goal-directed tasks and enabling the system to become more autonomous by learning skills from the human demonstrations.

## D.2 Learning for Task and Motion Planning

Task and Motion Planning (TAMP) is a powerful approach for solving challenging manipulation tasks by breaking them into smaller, easier to solve symbolic-continuous search problems [5, 22, 4, 23]. However, TAMP requires prior knowledge of skills and environment models, making it unsuitable for contact-rich tasks where hand-defining models is difficult. Recent works have proposed to learn environment dynamic models [24, 25, 26], skill operator models [27, 28], and skill samplers [29, 30]. However, these methods still require a complete set of hand-crafted skills. Closest to our work are LEAGUE [31] and Silver *et al.* [32] that learn TAMP-compatible skills. However, both works are limited in their real-world applicability. LEAGUE relies on hand-defined TAMP plan sampler and expensive RL procedures to learn skills in simulation, while Silver *et al.* requires hard-coded demonstration policies that can already solve the target tasks. Our work instead leverage human demonstrations to both train visuomotor skills and informing TAMP plan sampling. We empirically show that HITL-TAMP can efficiently solve challenging tasks such as making coffee in the real world.

## D.3 Imitation Learning from Human Demonstrations

Imitation learning techniques based on deep neural networks have shown remarkable performances in solving real-world manipulation tasks [6, 1, 10, 2, 7, 11]. We take a data-centric view [8, 2, 11] to scaling up imitation learning — HITL-TAMP speeds up demonstration collection for a wide range of contact-rich manipulation tasks. A trained HITL-TAMP also acts as a hierarchical policy [56]. The key difference to pure data-driven approaches [10, 56, 39, 8, 57] is that in HITL-TAMP, the TAMP framework directly drives the hierarchy to ensure that the learned skills are modular and compatible. Similarly, our work builds on research in combining learned and predefined skills [17, 18, 19, 20, 21] and formalizes human demonstrations and learned skills within a TAMP framework.

## E Tasks

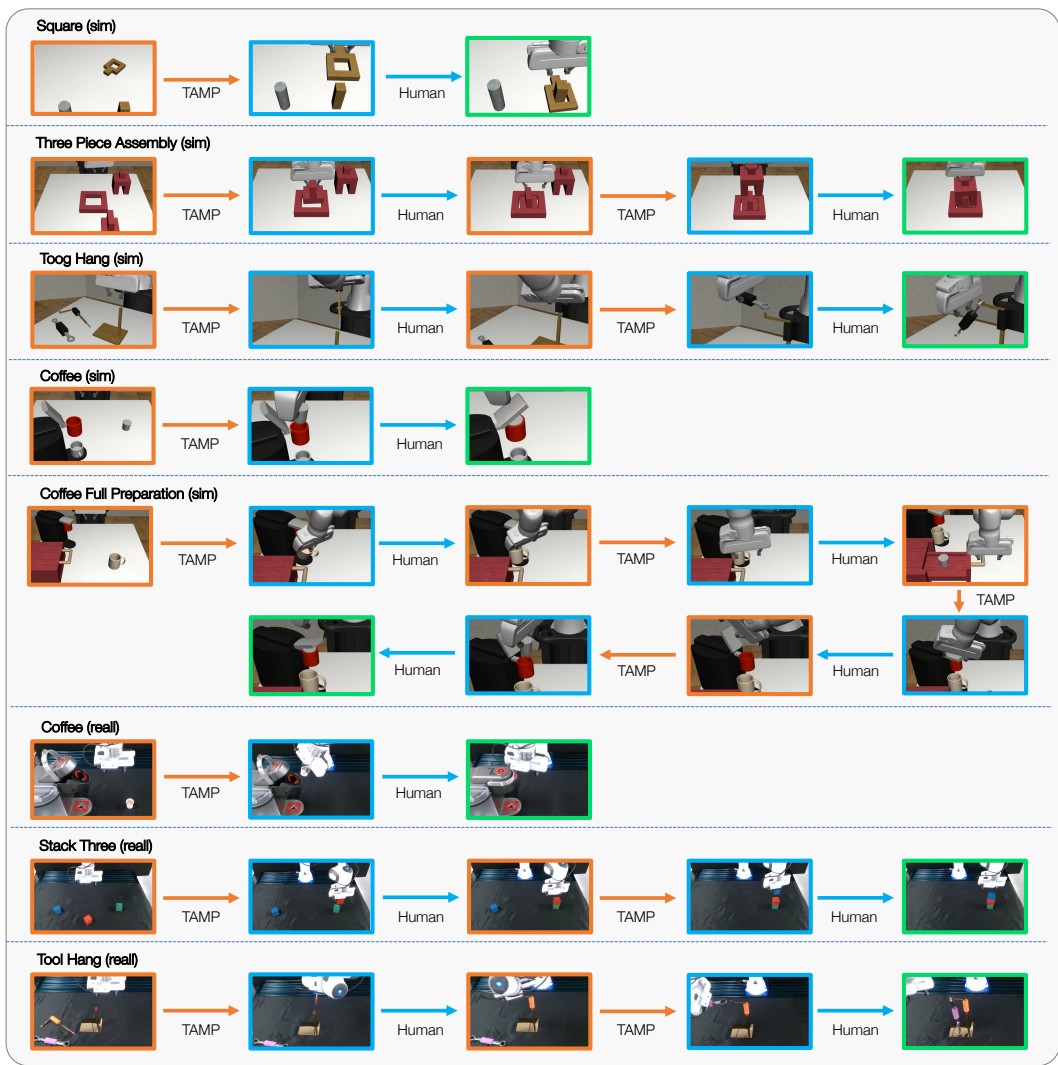

Figure E.1: **Task Segments.** We show the human and TAMP segments for each task.

In this section, we present extended task descriptions for each task, including a breakdown of which segments the human controls and which TAMP handles (see Fig. E.1).

**Stack Three (real).** The robot must stack 3 randomly placed cubes. The task consists of 4 total segments — TAMP handles grasping each cube and approaching the stack, and the human handles the placement of the 2 cubes on top of the stack.

**Square [58, 1] (sim).** The robot must pick a nut and place it onto a peg. The nut is initialized in a small region and the peg never moves. This task consists of two segments — TAMP grasps the nut and approaches the peg, and the human inserts the nut onto the peg.

**Square Broad (sim).:** The nut and peg are initialized anywhere on the table.

**Coffee [42] (sim + real).** The robot must pick a coffee pod, insert it into a coffee machine, and close the lid. The pod starts at a random location in a small, box-shaped region, and the machine is fixed. The task has two segments — TAMP grasps the pod and approaches the machine, and the human inserts the pod and closes the lid.

**Coffee Broad (sim + real).** The pod and the coffee machine have significantly larger initialization regions. With 50% probability, the pod is placed on the left of the table, and the machine on the right side, or vice-versa. Once a side is chosen for each, the machine location and pod location are further randomized in a significant region.

**Three Piece Assembly (sim).** The robot must assemble a structure by inserting one piece into a base and then placing a second piece on top of the first. The two pieces are placed around the base, but the base never moves. The tasks consists of four segments — TAMP grasps each piece and approaches the insertion point while the human handles each insertion.

**Three Piece Assembly Broad (sim).** The pieces are placed anywhere in the workspace.

**Tool Hang [1] (sim + real).** The robot must insert an L-shaped hook into a base piece to assemble a frame, and then hang a wrench off of the frame. The L-shaped hook and wrench vary slightly in pose, and the base piece never moves. The task has four segments — TAMP handles grasping the L-shaped hook and the wrench, and approaching the insertion / hang points, while the human handles the insertions.

**Tool Hang Broad (sim).** All three pieces move in larger regions of the workspace.

**Coffee Full Preparation (sim).** The robot must place a mug onto a coffee machine, retrieve a coffee pod from a drawer, insert the pod into the machine, and close the lid. The task has 8 segments — first TAMP grasps the mug and approaches the placement location, then the human places the mug on the coffee machine (the placement requires precision due to the arm size and space constraints). Next, TAMP approaches the machine lid, and the human opens the lid (requires extended contact with an articulated mechanism). Then, TAMP approaches the drawer handle, and the human opens the drawer. Finally, TAMP grasps the pod from inside the drawer and approaches the machine, and the human inserts the pod and closes the machine lid.

## F    Additional Data Throughput Comparisons

| Task | HITL-TAMP Time (min) | Conventional Time (min) |
|------|---------------------:|------------------------:|
| Square | **13.5** | 35.0 |
| Square Broad | **14.0** | 48.0 |
| Coffee | **22.6** | 46.4 |
| Coffee Broad | **28.8** | 57.8 |
| Tool Hang | **48.0** | 97.1 |
| Tool Hang Broad | **51.5** | 109.8 |
| Three Piece Assembly | **30.0** | 60.0 |
| Three Piece Assembly Broad | **34.9** | 68.3 |
| Coffee Preparation | **78.4** | 132.7 |
| **Total** | **321**.7 | 655.1 |

Table F.1: **Collection time comparison to conventional teleoperation datasets.** An extended comparison of data collection time for 200 demos across several tasks for both HITL-TAMP and the conventional teleoperation system. Some items were estimated using the time spent collecting 10 human demonstrations.

In this section, we compare how long it would have taken to collect our 2.1K+ HITL-TAMP demonstrations with a conventional teleoperation system. The results are shown in Table F.1. Several of the numbers were estimated by collecting 10 human demonstrations and multiplying by 20 (due to the time burden of collecting 200 human demonstrations across all tasks with a conventional teleoperation system). In most cases, HITL-TAMP takes more than 2x fewer minutes to collect 200 demos than the conventional system.

# G  Robustness to Pose Error

| Dataset | L0 | L1 | L2 |
|---|---|---|---|
| Square (L1) | $100.0 \pm 0.0$ | $100.0 \pm 0.0$ | $99.3 \pm 0.9$ |
| Square (L2) | $100.0 \pm 0.0$ | $100.0 \pm 0.0$ | $100.0 \pm 0.0$ |
| Coffee (L1) | $100.0 \pm 0.0$ | $100.0 \pm 0.0$ | $91.3 \pm 2.5$ |
| Coffee (L2) | $100.0 \pm 0.0$ | $99.3 \pm 0.9$ | $98.0 \pm 1.6$ |

Table G.1: **HITL-TAMP Robustness to Pose Noise.** We added uniform pose noise to all object poses perceived by our TAMP system. We use two levels of uniformly sampled noise - L1 is 5 mm of position noise and 5 degrees of rotation noise, and L2 is 10 mm of position noise and 10 degrees of rotation noise. For each level of noise, we collected 200 demonstrations with our HITL-TAMP system, trained image-based agents on these datasets, and evaluated the agents on the L0 (no noise), L1 noise, and L2 noise setting. The agents only perceive camera images and robot proprioception (*i.e.* not object poses), and the TAMP system receives noisy object poses. The results show that HITL-TAMP agents retain strong performance.

Since TAMP plans to pre-contact poses (constraints learned from human demos), errors in the hand-off location to the human operator are completely tolerable, as the human can account for any differences during their demonstration. Our real-world experiments in Fig. 5 best demonstrate the robustness of our system to pose error. For Coffee, we used an extremely crude box model of the coffee machine without any fine-grained pose registration. For ToolHang, the stand is not accurately captured in the observed point cloud due to the thinness of the stand base and column. Consequently, pose registration is naturally noisy. Despite these problems with perception, we were able to achieve high success rates in both tasks with few demonstrations.

In this section, we conduct an additional experiment in simulation to obtain quantitative evidence of HITL-TAMP's robustness to object pose estimation. We first describe our noise model. We added uniform pose noise to all object poses perceived by our TAMP system. We use two levels of uniformly sampled noise - L1 is 5 mm of position noise and 5 degrees of rotation noise and L2 is 10 mm of position noise and 10 degrees of rotation noise [59]. For each level of noise, we collected 200 demonstrations with our HITL-TAMP system (to be consistent with Fig. 6) on the Square and Coffee tasks, resulting in 4 new datasets in total. We then trained image-based agents on these datasets, and evaluated the agents on the L0 (no noise), L1 noise, and L2 noise setting. We emphasize that the agents only perceive camera images and robot proprioception, not object poses, and the TAMP system receives noisy object poses.

The results are presented in Table G.1. Each row corresponds to agents trained on one of our new datasets and each column corresponds to different levels of noise applied to the TAMP system during policy evaluation. Recall that we report the success rates across 50 evaluations, where there are no TAMP failures, and 3 seeds (discussed further in Appendix K and Appendix M).

When evaluating the agents trained on the L1 and L2 datasets on the same levels, the results are near-perfect (100% success rate for all except Coffee L2, which gets 98% success), which aligns with the 100% success achieved by our agents on our noise-free datasets (see Fig. 6, left). We also found that training on higher amounts of noise gives our trained agents some level of robustness to lower amounts of noise (*e.g.* evaluating the L2 models on L0 and L2).

We also analyze the execution failure rate of the TAMP system itself (which corresponds to how often we terminate an episode due to a failed grasp of the nut / coffee pod, or dropping the object in hand). We found that the TAMP system failure rate increases by level: from 0% on L0 to 6% on L1 and 23% on L2 for Square, and from 0% on L0 to 6% on L1 and 24% on L2 for Coffee. This is to be expected, as erroneous poses can lead to a bad grasp. In such settings where perception errors to cause grasp failures, we could easily have the human teleoperate the grasping part of each trajectory as well during data collection and then have the trained agent learn that task segment as well.

 **H   Demonstration Statistics**

| Task | Human | Trajectory (HT) | Trajectory (C) |
|------|-------|-----------------|----------------|
| Square | 19.8 | 582.2 | 150.8 |
| Square Broad | 24.2 | 647.8 | 167.9 |
| Coffee | 71.6 | 472.0 | 199.3 |
| Coffee Broad | 90.6 | 663.7 | 273.8 |
| Tool Hang | 70.4 | 1297.9 | 479.8 |
| Tool Hang Broad | 71.3 | 1485.8 | 522.6 |
| Three Piece Assembly | 35.3 | 897.9 | 260.1 |
| Three Piece Assembly Broad | 39.6 | 1174.1 | 342.0 |
| Coffee Preparation | 43.8 | 1328.6 | 593.2 |
| Stack Three (real) | 60.9 | 499.2 | - |
| Coffee (real) | 295.3 | 494.9 | - |
| Coffee Broad (real) | 326.5 | 548.3 | - |
| Tool Hang (real) | 124.3 | 1144.5 | - |

Table H.1: **Demonstration Lengths.** For each task, we report the average length (time steps) of the human segment, the average trajectory length of our HITL-TAMP datasets (HT), and as a point of comparison, the average trajectory length of the conventional system data (C). Note that if a trajectory contains multiple human segments, we average them.

In Table H.1, we present the average length (time steps) of the human-provided segment, the average trajectory length of our HITL-TAMP datasets (HT), and as a point of comparison, the average trajectory length of the conventional system data (C). Note that if a trajectory contains multiple human segments, we average across them, and that some of the conventional system lengths are estimates based on collecting 10 trajectories (the same ones used for the analysis in Appendix F). We see that the average human segment is small compared to the entire trajectory length — this might help explain the efficacy of our TAMP-gated policy, since the policy is only responsible for short-horizon, contact-rich behaviors.

## I  Queueing System Analysis

In Sec. 4 and Fig. 3, we discussed our queueing system, which enables scalable data collection with HITL-TAMP by allowing a single human operator to manage a fleet of $N_{\mathrm{robot}}$ robot arms and ensuring that the human operator is always kept busy. In this section, we provide some additional derivations and analysis on how the choice of the number of robot arms influences data throughput.

Assuming that the human has an average queue consumption rate (number of task demonstrations completed per unit time) of $R_H$ and the TAMP system has an average queue production rate (number of task segments executed successfully per unit time) of $R_T$, we would like the effective rate of production to match or exceed the rate of consumption,

$$R_T(N_{\mathrm{robot}} - 1) \geq R_H.$$

Here, the minus 1 is because 1 robot is controlled by the human. Rearranging, we obtain $N_{\mathrm{robot}} \geq 1 + \frac{R_H}{R_T}$. Thus, the size of the fleet should be at least one more than the ratio between the human rate of producing demonstration segments and the TAMP rate of solving and executing segments.

This number is often limited by either the amount of system resources (in simulation) or the availability of hardware (in real world). In practice, human operators also need to take breaks and have an effective "duty cycle" where they are kept busy $X\%$ of the time. HITL-TAMP can support this extension as well. Assume that the human is operating the system for $T_{\mathrm{on}}$ and resting for $T_{\mathrm{off}}$. The human consumes items in the queue during $T_{\mathrm{on}}$ at an effective rate of

$$R_H - R_T(N_{\mathrm{robot}} - 1),$$

and has the queue filled up during $T_{\mathrm{off}}$ at a rate of $R_T(N_{\mathrm{robot}} - 1)$. Ensuring that the human consumption rate is less than or equal to the production rate, we have

$$T_{\mathrm{on}}(R_H - R_T(N_{\mathrm{robot}} - 1)) \leq T_{\mathrm{off}} R_T(N_{\mathrm{robot}} - 1).$$

After rearranging we arrive at

$$N_{\mathrm{robot}} \geq 1 + \frac{R_H}{R_T} \frac{X}{100},$$

where

$$\frac{X}{100} = \frac{T_{\mathrm{on}}}{(T_{\mathrm{on}} + T_{\mathrm{off}})}$$

is the human duty cycle ratio.

## J  Additional Details on TAMP-Gated Teleoperation

We provide additional details on how TAMP-gated teleoperation works. The TAMP system decides when to execute portions of a task, and when a human operator should complete a portion. Each teleoperation episode consists of one or more *handoffs* where the TAMP system prompts a human operator to control a portion of a task, or where the TAMP system takes control back after it determines that the human has completed their segment.

Algorithm 1 displays the pseudocode of the HITL-TAMP system: TAMP-GATED-CONTROL. It takes as input goal formula $G$. On each TAMP iteration, it observes the current state $s$. If it satisfies the goal, the episode terminates successfully. Otherwise, the TAMP system solves for a plan $\vec{a}$ using PLAN-TAMP from current state $s$ to the goal $G$. We implement PLAN-TAMP using the *adaptive* PDDLStream algorithm [23]. The TAMP system then deploys its controller EXECUTE-JOINT-COMMANDS and issues joint position commands to the robot to carry out planned motions until reaching an action $a$ that requires the human. At this time, control switches into teleoperation mode, where the human has full 6-DoF control of the end effector. We use a smartphone interface and map phone pose displacements to end effector displacements, similar to prior teleoperation systems [36, 37, 10]. The robot end effector is controlled using an Operational Space Controller [38]. As in [42], we apply phone pose differences as relative pose commands to the current end effector pose. This allows control to be decoupled from the current configuration of the robot arm, which is important as the TAMP system can prompt the human to takeover in diverse configurations. While the human is controlling the robot, the TAMP system monitors whether the state satisfies the planned action postconditions $a.effects$. Once satisfied, control switches back to the TAMP system, which replans.

---

**Algorithm 1** TAMP-Gated Teleoperation

---

1: **procedure** TAMP-GATED-CONTROL($G$)
2:   **while True do**
3:     $s \leftarrow$ OBSERVE()                                               ▷ Estimate or observe state
4:     **if** $s \in G$ **then**                                             ▷ State satisfies goal
5:       **return True**                                                   ▷ Success!
6:     $\vec{a} \leftarrow$ PLAN-TAMP($s, G$)                                ▷ Solve for a plan $\vec{a}$
7:     **for** $a \in \vec{a}$ **do**                                       ▷ Iterate over actions
8:       **if not** IS-HUMAN-ACTION($a$) **then**
9:         EXECUTE-JOINT-COMMANDS($a$)
10:       **else**
11:         **while** OBSERVE() $\notin a.$**eff do**
12:           EXECUTE-TELEOP()                                              ▷ Teleoperation
13:         **break**                                                       ▷ Re-observe and re-plan

---

### J.1  Example Plan

Consider a plan found by the TAMP system for the **Tool Hang** task on the first planning invocation:

$$\vec{a}_1 = [\texttt{move}(\boldsymbol{q_0}, \tau_1, q_1), \texttt{pick}(frame, g^f, \boldsymbol{p_0^f}, q_1), \texttt{move}(q_1, \tau_2, q_2), \underline{\texttt{attach}(frame, g^f, p_2, q_2, \widehat{p}_2^f, \widehat{q}_2, stand)},$$
$$\texttt{move}(\widehat{q}_2, \widehat{\tau}_3, q_3), \texttt{pick}(tool, g^t, \boldsymbol{p_0^t}, q_3), \texttt{move}(q_3, \tau_4, q_4), \texttt{attach}(tool, g^t, p_4, q_4, \widehat{p}_4^t, \widehat{q}_4, frame)].$$

The values in bold represent constants present in the initial state; the non-bold values are parameter values selected by the planner. The learned preimages enable the TAMP system to plan not only a trajectory $\tau_1$ to the first manipulation but also to the second manipulation $\tau_2$. However, because the third trajectory $\widehat{\tau}_3$ depends on the resultant configuration $\widehat{q}_2$, planning for it is deferred. Upon successfully achieving $\texttt{Attached}(frame, stand)$, replanning produces a new plan.

## K  Policy Training Details

In this section, we detail how we train policies via imitation learning from the human segments of HITL-TAMP datasets. Many choices are mirrored from Mandlekar *et al.* [1].

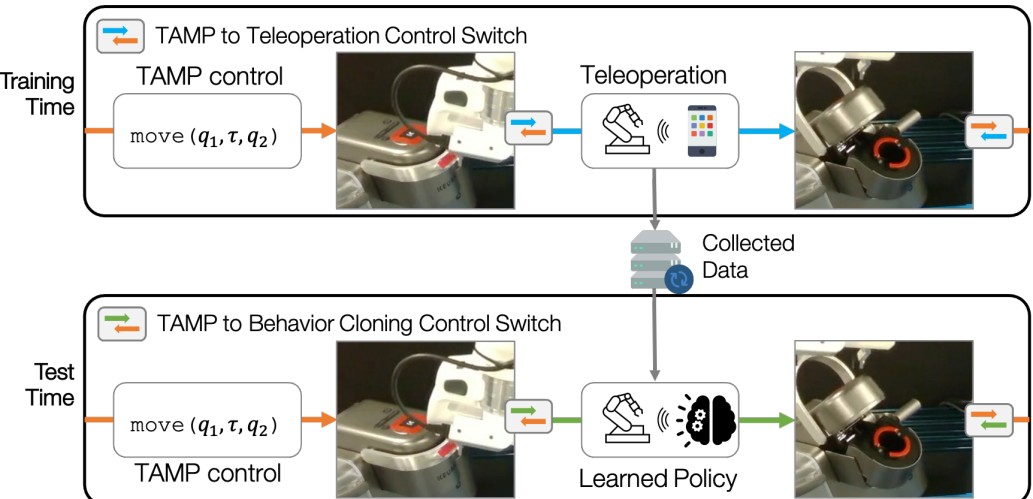

Figure K.1: **Training and Testing Policies.** The top row shows the HITL-TAMP policy at training time, where a human teleoperate certain segments, such as opening the coffee machine lid. The bottom row shows the HITL-TAMP policy at testing time, where the human segments are replaced with a learned policy trained with behavior cloning using the collected training data.

### K.1  Observation Spaces

In our experiments, policies are either trained on low-dim state observations or image observations — this kind of flexibility is advantageous as it eases the burden of perception for deploying TAMP systems in the real world. Low-dim observations include ground-truth object poses, while image observations consist of RGB images from a front-view camera and a wrist-mounted camera. Both observations include proprioception (end-effector pose and gripper finger width). In simulation, the image resolution is 84x84, while in real world tasks, we use a resolution of 120x160 for Stack Three, Coffee, and Coffee Broad, and a resolution of 240x240 for Tool Hang. Our real-world agents are all image-based, since we do not assume that objects can be tracked. The real-world Tool Hang agent did not use the wrist-view in observations, since we found that it was completely occluded during the human portions of the task. The TAMP system only estimates poses at the start of each episode. We use a simple perception pipeline consisting of RANSAC plane estimation to segment the table from the point cloud, DBSCAN [60] to cluster objects, color-based statistics to associate objects, and Iterative Closest Point (ICP) to estimate object poses. For image-based agents, we apply pixel shift randomization (up to 10% of each image dimension) as a data augmentation technique (as in Mandlekar *et al.* [1]).

### K.2  Action Space

As described in Sec. 3.3, we collect training data using teleoperation through 6-DoF end-effector control, where an Operational Space Controller [38] interprets delta end-effector actions and converts them to joint commands. Thus, the action space for all policy learning is also 6-DoF end-effector poses.

### K.3  Training and Evaluation

We use BC-RNN with default hyperparameters from Mandlekar *et al.* [1] with the exception of an increased learning rate of $10^{-3}$ for policies trained on low-dim observations, to train policies from

the human segments in each dataset. We follow the policy evaluation convention from Mandlekar *et al.* [1], and report the maximum Success Rate (SR) across all checkpoint evaluations over 3 seeds, which is evaluated over 50 rollouts. However, the TAMP system can fail during a rollout. To decouple TAMP failures from policy failures, we keep conducting rollouts for each checkpoint until 50 rollouts with no TAMP failures have been collected, and compute policy success rate over those rollouts (discussion in Appendix M). In the real world, we take the final policy checkpoint from training, and use it for evaluation. Fig. K.1 visualizes the difference between the HITL-TAMP policy at training time, where teleoperation is used, and at testing time, where teleoperation is substituted with a learned policy for fully autonomous control.

# L  Low-Dim Policy Training Results

| Task | Time (min) | SR (im) | TAMP-gated SR (im) |
|------|-----------:|--------:|-------------------:|
| Square (C) | 25.0 | $84.0 \pm 0.0$ | $91.3 \pm 5.2$ |
| Square (HT) | **13.5** | **$100.0 \pm 0.0$** | **$100.0 \pm 0.0$** |
| Square Broad (C) | 48.0 | $29.3 \pm 0.0$ | $88.0 \pm 1.6$ |
| Square Broad (HT) | **14.0** | **$100.0 \pm 0.0$** | **$100.0 \pm 0.0$** |
| Three Piece Assembly (C) | 60.0 | $55.3 \pm 0.0$ | **$96.0 \pm 2.8$** |
| Three Piece Assembly (HT) | **30.0** | **$100.0 \pm 0.0$** | **$100.0 \pm 0.0$** |
| Tool Hang (C) | 80.0 | $29.3 \pm 0.0$ | $60.0 \pm 19.6$ |
| Tool Hang (HT) | **48.0** | **$80.7 \pm 1.9$** | **$80.7 \pm 1.9$** |

Table L.1: **Comparison to conventional teleoperation datasets (low-dim).** We trained normal and TAMP-gated policies using conventional teleoperation (C) and compared them to HITL-TAMP (HT). TAMP-gating makes policies trained on the data comparable to HITL-TAMP data, but data collection still involves significantly higher operator time.

In Table 6 and Sec. 6.2, we only presented results with image policies. In this section, we show that HITL-TAMP still compares favorably to conventional teleoperation data when trained on low-dim observations. The results are presented in Table L.1.

## M   TAMP Success Analysis

| Task | Time (min) | SR (low-dim) | SR (image) | TAMP SR (low-dim) | Raw SR (low-dim) | TAMP SR (image) | Raw SR (image) |
|---|---|---|---|---|---|---|---|
| Square | 13.5 | $100.0 \pm 0.0$ | $100.0 \pm 0.0$ | $77.7 \pm 1.5$ | $77.7 \pm 1.5$ | $82.0 \pm 1.9$ | $82.0 \pm 1.9$ |
| Square Broad | 14.0 | $100.0 \pm 0.0$ | $100.0 \pm 0.0$ | $81.2 \pm 2.7$ | $81.2 \pm 2.7$ | $76.1 \pm 5.1$ | $76.1 \pm 5.1$ |
| Coffee | 22.6 | $100.0 \pm 0.0$ | $100.0 \pm 0.0$ | $100.0 \pm 0.0$ | $100.0 \pm 0.0$ | $100.0 \pm 0.0$ | $100.0 \pm 0.0$ |
| Coffee Broad | 28.8 | $99.3 \pm 0.9$ | $96.7 \pm 0.9$ | $98.1 \pm 1.6$ | $97.4 \pm 0.9$ | $97.4 \pm 0.9$ | $94.2 \pm 0.1$ |
| Tool Hang | 48.0 | $80.7 \pm 1.9$ | $78.7 \pm 0.9$ | $97.4 \pm 1.8$ | $78.6 \pm 2.9$ | $97.4 \pm 1.8$ | $76.6 \pm 1.2$ |
| Tool Hang Broad | 51.5 | $49.3 \pm 1.9$ | $40.7 \pm 0.9$ | $88.8 \pm 1.9$ | $43.8 \pm 0.8$ | $93.8 \pm 0.8$ | $38.1 \pm 1.1$ |
| Three Piece Assembly | 30.0 | $100.0 \pm 0.0$ | $100.0 \pm 0.0$ | $96.2 \pm 1.5$ | $96.2 \pm 1.5$ | $95.0 \pm 2.3$ | $95.0 \pm 2.3$ |
| Three Piece Assembly Broad | 34.9 | $84.7 \pm 4.1$ | $82.0 \pm 1.6$ | $71.4 \pm 0.0$ | $60.5 \pm 2.9$ | $76.0 \pm 4.0$ | $62.3 \pm 4.3$ |
| Coffee Preparation | 78.4 | $96.0 \pm 3.3$ | $100.0 \pm 0.0$ | $80.9 \pm 4.8$ | $77.6 \pm 4.4$ | $83.8 \pm 1.8$ | $83.8 \pm 1.8$ |

Table M.1: **Analyzing TAMP Success Rates during Policy Evaluations.** A more complete set of results from Table 6 on HITL-TAMP datasets to demonstrate that policy evaluations do not have significant bias by only evaluating in regions where TAMP is successful. All TAMP success rates are high (above 70%) and most are above 88%.

Recall that when evaluating a trained policy, to decouple TAMP failures from policy failures, we keep conducting rollouts for each checkpoint until 50 rollouts with no TAMP failures have been collected, and compute policy success rate over those rollouts. In certain cases, this procedure could lead to biased evaluations — for example, if TAMP is only successful for an object in a limited region of the robot workspace. In this section, we present the TAMP success rates and raw success rates (including TAMP failures) for the policies in Table 6 (left), and demonstrate that it is unlikely that such bias exists in our evaluations. We present the results in Table M.1 — note that the Time and SR columns are reproduced from Table 6 (right) for ease of comparison. We see that all TAMP success rates are high (above 70%) and most are above 88%.

# N   Additional Details on Conventional Teleoperation System

In this section, we provide additional details on the conventional teleoperation system that we compared against in this work (*e.g.* in Table 1 and Fig. 6) as well as explain why it is a representative baseline to compare against. Prior works in imitation learning leveraged robot teleoperation systems to allow for full 6-DoF control of a robot manipulator. These systems typically map the state of a teleoperation device, such as a Virtual Reality controller [6], a 3D mouse [61], a smartphone [36, 37], or a point-and-click web interface [9], to a desired robot end effector pose. They also use an end-effector controller to try and achieve the desired pose specified by the teleoperation device. The operator controls the robot arm in real-time by using the teleoperation device.

This teleoperation paradigm has been used extensively in prior work that collects and learns from human demonstrations [1, 6, 2, 7, 41, 20, 9, 36, 37, 42, 10, 15, 16, 11]. In this work, we compared against the RoboTurk [36, 37] system and smartphone interface, which has been used in several prior imitation learning works [10, 39, 1, 62, 63, 64]. It was also used to collect datasets for the robomimic benchmark [1], whose results we also compare against (see Sec. 6.2). This makes it an appropriate baseline. However, it is important to note that our HITL-TAMP system is not specific to a particular teleoperation interface – in fact, our system is also compatible with a 3D mouse interface [61].

## O  Additional User Study Details

In this section, we provide additional details on how the user study was conducted. We recruited 15 participants that had varying levels of experience with robot teleoperation: 4 participants were unfamiliar, 6 were somewhat familiar, and 5 were very familiar with it. The purpose of the study was to compare our system (HITL-TAMP) to a conventional teleoperation system [36], where task demonstrations were collected without TAMP involvement. Participants underwent a brief tutorial (5-10 minutes) to familiarize themselves with the smartphone teleoperation interface and to practice collecting task demonstrations using both systems.

Each participant performed task demonstrations on 3 tasks (**Coffee**, **Square (Broad)**, and **Three Piece Assembly (Broad)**) for 10 minutes on each system, totaling 60 minutes of data collection across the 3 tasks and 2 systems. To reduce bias, the order of systems was randomized for each task and user (while maintaining the task order). Participants filled out a post-study survey to rank their experience with both systems. Each participant's number of successful demonstrations was recorded to evaluate the data throughput of each system, and agents were trained on each participant's demonstrations and across all participants' demonstrations (Sec. 6.1). See Appendix K for full details on policy training.

All demonstrations were collected on a single workstation with an NVIDIA GeForce RTX3090 GPU. We used 6 robot processes ($N_{\text{robot}} = 6$) to ensure that human operators were always kept busy (see Sec. 4).

