# OpenReview forum: "Human-in-the-Loop Task and Motion Planning for Imitation Learning"
_robot-learning.org/CoRL/2023/Workshop/TGR — CoRL 2023 Workshop TGR Poster_

### Official Review · Reviewer_S2Bt · 2023-10-19

**Rating:** 8
**Confidence:** 5

**Review:**

The proposed methods combines human teleoperation and TAMP, thus addressing limitations for generation using TAMP. The method is sound and very relevant to the workshop topic.

---

### Official Review · Reviewer_Yacg · 2023-10-20

**Rating:** 8
**Confidence:** 4

**Review:**

This paper presents an efficient method that enhances the process of human demonstration collection by integrating Task and Motion Planning (TAMP) with human demonstrations.

---

### Decision · Program_Chairs · 2023-10-20

**Decision:**

Accept (Poster)

**Comment:**

Great paper and closely aligned topic!